# Serological Evidence for Circulation of Influenza D Virus in the Ovine Population in Italy

**DOI:** 10.3390/pathogens13020162

**Published:** 2024-02-11

**Authors:** Gianvito Lanave, Michele Camero, Chiara Coppola, Serena Marchi, Giuseppe Cascone, Felice Salina, Miriana Coltraro, Amienwanlen E. Odigie, Emanuele Montomoli, Chiara Chiapponi, Vincenzo Cicirelli, Vito Martella, Claudia M. Trombetta

**Affiliations:** 1Department of Veterinary Medicine, University Aldo Moro of Bari, 70010 Valenzano, Italy; michele.camero@uniba.it (M.C.); amienwanlen.odigie@uniba.it (A.E.O.); vincenzo.cicirelli@uniba.it (V.C.); vito.martella@uniba.it (V.M.); 2Department of Molecular and Developmental Medicine, University of Siena, 53100 Siena, Italy; chiara.coppola2@unisi.it (C.C.); serena.marchi2@unisi.it (S.M.); emanuele.montomoli@unisi.it (E.M.); trombetta@unisi.it (C.M.T.); 3Istituto Zooprofilattico Sperimentale della Sicilia “A. Mirri”, 90129 Palermo, Italy; giuseppe.cascone@izssicilia.it (G.C.); felice.salina@izssicilia.it (F.S.); miriana.coltraro@gmail.com (M.C.); 4VisMederi S.r.l., 53035 Monteriggioni, Italy; 5OIE Reference Laboratory for Swine Influenza, Sede Territoriale di Parma, Istituto Zooprofilattico Sperimentale della Lombardia ed Emilia Romagna, 25124 Brescia, Italy; chiara.chiapponi@izsler.it

**Keywords:** influenza D virus, seroprevalence, hemagglutination inhibition assay, virus neutralization assay, ovine, Italy

## Abstract

Influenza D virus (IDV) is a novel orthomyxovirus initially isolated from pigs exhibiting influenza-like disease in the USA. Since then, IDV has been detected worldwide in several host species, including livestock animals, whilst specific antibodies have been identified in humans, raising concerns about interspecies transmission and zoonotic risks. Few data regarding the seroprevalence of IDV in small ruminants have been available to date. In this study, we assessed the prevalence of antibodies against IDV in ovine serum samples in Sicily, Southern Italy. Six hundred serum samples, collected from dairy sheep herds located in Sicily in 2022, were tested by haemagglutination inhibition (HI) and virus neutralization (VN) assays using reference strains, D/660 and D/OK, representative of two distinct IDV lineages circulating in Italy. Out of 600 tested samples, 168 (28.0%) tested positive to either IDV strain D/660 or D/OK or to both by HI whilst 378 (63.0%) tested positive to either IDV strain D/660 or D/OK or to both by VN. Overall, our findings demonstrate that IDV circulates in ovine dairy herds in Sicily. Since IDV seems to have a broad host range and it has zoonotic potential, it is important to collect epidemiological information on susceptible species.

## 1. Introduction

The family *Orthomyxoviridae* includes four types of influenza viruses, A, B, C, and D (IAV to IDV). IAV, IBV, and ICV are known to cause human respiratory disease and have also been described in several animal hosts [1]. IDV was first isolated from pigs with severe respiratory signs in the United States in 2011 [2]. IDV shares 50% of genetic identity with human ICV, although cross-reactivity with antibodies (Abs) directed against human ICV has not been observed so far [2]. The virus belongs to the genus *Deltainfluenzavirus* of the family *Orthomyxoviridae* according to the classification of the International Committee on Taxonomy of Viruses (ICTV) [3]. Reassortment of IDV with the other genera of influenza viruses has not been observed [3] and a possible IDV derivation from ICV has been suggested [4].

The presence of IDV has been reported globally, either directly or indirectly, in livestock animals [5], feral pigs and wild boars [6,7], camelids [8], small ruminants [9,10,11], and horses [12]. Abs specific to IDV have been also identified in humans [13], chiefly in workers with occupational exposure, i.e., veterinarians and farmers [14,15]. Although IDV was first isolated from pigs, cattle are presently considered as IDV’s main host and reservoir [16]. 

Several IDV lineages have been described in North America, Europe, and Asia and they have been designated as D/swine/Oklahoma/1334/2011 (D/OK) [2], D/bovine/Oklahoma/660/2013 (D/660) [3], D/bovine/Yamagata/10710/2016 [17], D/bovine/Yamagata/1/2019 [8], and D/bovine/California/0363/2019 [18] lineages. Recently, an additional lineage, D/bovine/Turkey-Bursa/ET-130/2013, has been tentatively proposed, although phylogenetic analysis relied solely on a partial HEF gene sequence [19]. D/OK and D/660 are regarded as the two major circulating lineages in North America and Europe with some reassortment events between strains of these lineages being described [20,21]. In Italy, D/OK and D/660 lineages co-circulate in the cattle population [20].

IDV can replicate in nasal, tracheal, and lung ovine tissues, used as an ex vivo model, indicating an average susceptibility of this host species to IDV infection [22]. The virus circulates in small ruminants with an observed seroprevalence generally lower than in cattle, camels, or pigs [9,10,11,23,24,25]. Viral RNA has also been identified in small ruminants [26]. The limited data available in the ovine population hinder the depiction of a clear picture of IDV circulation. 

In Sicily, the ovine dairy sector represents a significant economic resource, chiefly in marginal rural areas, and Sicily is the second highest Italian region for the number of sheep reared for milk production. The native dairy breeds (Valle del Belìce, Comisana, Barbaresca, and Pinzirita) are well adapted to produce high-quality milk and dairy products in difficult climatic conditions and harsh environments, thus providing a unique ecosystem. The aim of this study was to assess the prevalence of Abs against IDV in the ovine population in Sicily, Italy.

## 2. Materials and Methods

### 2.1. Study Area

The study was carried out in southeastern Sicily, Italy. In Sicily, the ovine dairy sector constitutes a significant economic resource, chiefly for those peripheral rural areas in which alternative economic activities are hindered by environmental constraints. After Sardinia, Sicily is regarded as the second highest Italian region for the number of sheep raised for milk production, counting about 699,000 animals (data provided by BDN–Anagrafe Zootecnica) [27]. 

### 2.2. Reference Viruses

Influenza D/bovine/Oklahoma/660/2013 virus (D/660 lineage, hereby referred to as D/660), kindly provided by Prof. Feng Li, University of Kentucky, and influenza D/swine/Italy/199724-3/2015 virus (D/OK lineage, hereby referred to as D/OK), obtained from the European Virus Archive (EVAg), were propagated in Madin-Darby Canine Kidney (MDCK) cells, as previously described [13].

### 2.3. Serum Samples

A total of 600 serum samples were collected from 15 dairy sheep herds of Comisana breed, located in two prefectures (Ragusa and Syracuse) of Sicily region from March to May 2022. 

Four hundred and seven (range 10 to 70) samples were collected from 9 dairy sheep herds located in Ragusa prefecture whilst 193 (range 7 to 57) samples were collected from 6 dairy sheep herds located in Syracuse prefecture (Table 1). All the animals were sampled by the Istituto Zooprofilattico Sperimentale della Sicilia in the frame of national plans for brucellosis and they were more than 6 months old. 

Influenza D hyperimmune pig serum to D/swine/Italy/199724-3/2015 was used as positive control (IZSLER, Brescia, Italy).

Human serum without immunoglobulin A, immunoglobulin M, and immunoglobulin G was used as a negative control (Sigma-Aldrich, St. Louis, MO, USA).

All serum samples were tested by haemagglutination inhibition (HI) and virus neutralization (VN) assays.

### 2.4. HI Assay

The HI assay was performed as previously described [13]. All serum samples, including positive and negative controls, were pre-treated with receptor-destroying enzyme (RDE) from Vibrio cholerae (ratio 1:4) (Denka, Tokyo, Japan) for 18 h at 37 °C in a water bath and then heat inactivation for 1 h at 56 °C. Before testing, the samples were re-treated with 15% turkey red blood cells (RBCs) (ratio 1:1, final sample dilution 1:10) and tested in triplicate using turkey RBCs adjusted to a final dilution of 0.35%. The antibody titre was expressed as the reciprocal of the highest serum dilution showing complete inhibition of agglutination. Since the starting dilution was 1:10, titres below the detectable threshold were conventionally expressed as 5 for calculation purposes (half the lowest detection threshold).

### 2.5. VN Assay

The VN assay was performed as previously described [13]. The MDCK cell cultures were grown at 37 °C in 5% CO_2_ and pre-incubated in a 96-well plate for 4 h. Serum samples and positive and negative controls were previously heat-inactivated at 56 °C for 30 min, twofold diluted with UltraMDCK culture medium (Lonza, Walkersville, MD, USA), and then mixed with an equal volume of virus (100 TCID_50_/well). After 1 h of incubation at 37 °C in 5% CO_2_, 100 µL of the mixture was added to the MDCK cell suspension. Plates were read for virus neutralization activity in the supernatant after 4 days of incubation at 37 °C in 5% CO_2_. VN assays were performed in triplicate. The VN titre was expressed as the reciprocal of the highest serum dilution showing the absence of virus neutralization.

### 2.6. Data Analysis

Discrete data (results from HI and VN assays), defined as new categorical dichotomous variables, were described as counts and percentages and compared between different reference IDV strains and prefectures by a χ2 test. Calculation of the 95% confidence intervals for the proportions was based on the Clopper–Pearson exact method [28]. Statistical analyses were performed by using the software package EZR version 1.40 (Saitama Medical Centre, Jichi Medical University, Saitama, Japan) [29]. *p* < 0.05 was considered as statistically significant.

## 3. Results

To assess if the ovine population in Italy is susceptible to IDV, we tested 600 dairy sheep serum samples gathered in 2022 from 15 herds located in two prefectures (Ragusa and Syracuse) of the Sicily region (Figure 1).

Influenza D pig serum hyperimmune to D/swine/Italy/199724-3/2015 (D/OK lineage) was used as a positive control and it showed HI titres of 2560 to an homologous strain and 640 to the D/bovine/Oklahoma/660/2013 virus (D/660 lineage).

All samples were tested for specific antibodies against two different IDV reference strains, D/660 and D/OK, using HI and VN assays, repeated in triplicate. Any serum sample with an average antibody titre greater than or equal to 10 was considered positive for Abs against the tested viral strain [30].

In the HI assay, 157 (26.2%, 95%CI/22.7–29.9, range 2 to 29) samples were positive for D/660 and 33 samples (5.5%, 95% CI/3.8–7.6, range: 1 to 5) were positive for D/OK (Table 2 and Appendix A). The comparison of the HI results between the two reference strains was statistically significant (*p* < 0.00001).

Abs reactive against D/660 were detected by HI in 119 (29.2%, 95% CI: 24.9–33.9, range: 2 to 29) out of 407 total samples from 9 out 9 dairy sheep herds located in the Ragusa prefecture, and in 38 (19.7%, 95% CI: 14.3–26.0, range: 3 to 20) out of 193 samples from 4 out 6 dairy sheep herds located in the Syracuse prefecture. The comparison of the results between the two prefectures displayed statistical significance (*p* = 0.017). A breakdown of the Abs titres of the 157 samples positive to D/660 by HI assay is as follows: 1/10 to 1/19 Abs titres in 83 (52.9%) samples, 1/20 to 1/39 in 55 (35.0%) samples, 1/40 to 1/79 in 13 (8.3%) samples, 1/80 to 1/159 in 3 (1.9%) samples, and 1/160 to 1/319 in 3 (1.9%) samples (Figure 2).

Abs reactive with D/OK were found in 20 (4.9%, 95% CI/3.0–7.5, range/1 to 5) out of 407 samples from 8 out of 9 sheep herds in the Ragusa prefecture, and from 13 (6.7%, 95% CI/3.6–11.2, range 2 to 5) out of 193 samples from 4 out of 6 sheep herds located in the Syracuse prefecture (Table 2). The comparison of the results between the two prefectures did not exhibit any statistical significance (*p* > 0.05). The 33 samples positive to D/OK encompassed 24 (72.7%) sera with a range of 1/10 to 1/19 Ab titres, 4 (12.1%) with a range of 1/20 to 1/39, 4 (12.1%) with a range of 1/40 to 1/79, and 1 (3.1%) with a range of 1/80 to 1/159 (Figure 2).

The ovine samples collected in this study were also screened using a VN assay. A total of 365 sera (60.8%, 95% CI: 56.8–64.8, range: 6 to 43) tested positive for D/660 whilst 128 samples (21.3%, 95% CI: 18.1–24.8, range: 4 to 23) were positive for D/OK (Table 2 and Appendix A). The comparison of the VN results between the two reference strains was statistically significant (*p* < 0.00001). Neutralizing Abs for D/660 were detected in 243 (59.7%, 95% CI: 54.8–64.5, range: 8 to 41) out of 407 samples from 9 out 9 dairy sheep herds of the Ragusa prefecture and in 122 (63.2%, 95% CI: 56.0–70.0, range: 6 to 43) out of 193 samples from 6 out of 6 dairy sheep herds located in the Syracuse prefecture. The comparison of the results between the two prefectures was statistically significant (*p* = 0.00006). A breakdown of the Abs titres of the 365 samples positive to D/660 by VN assay is as follows: 1/10 to 1/19 Ab titres in 44 (12.1%) samples, 1/20 to 1/39 in 127 (34.8%) samples, 1/40 to 1/79 in 123 (33.7%) samples, 1/80 to 1/159 in 42 (11.6%) samples, 1/160 to 1/319 in 21 (5.8%) samples, 1/320 to 1/639 in 6 (1.6%) samples, 1/640 to 1/1279 in 1 (0.2%) sample, and over 1280 in 1 (0.2%) sample (Figure 2).

Neutralizing Abs for D/OK were found in 76 (18.7%, 95% CI: 15.0–22.8, range: 5 to 16) out of 407 samples from 9 out of 9 sheep herds of the Ragusa prefecture and in 52 (26.9%, 95% CI: 20.8–33.8, range: 5 to 23) out of 193 samples from 5 out of 6 sheep herds located in the Syracuse prefecture. The comparison of the results between the two prefectures was not statistically significant (*p* > 0.05). A breakdown of the Abs titres in the 128 samples positive to D/OK by VN is as follows: 23 samples (18.0%) displayed a range of 1/10 to 1/19 Ab titres, 52 (40.6%) a range of 1/20 to 1/39, 36 (28.1%) a range of 1/40 to 1/79, 9 (7.0%) a range of 1/80 to 1/159, 4 (3.1%) a range of 1/160 to 1/319, 3 (2.3%) a range of 1/320 to 1/639, and 1 (0.9%) a range of 1/640 to 1/1280 (Figure 2).

Reactivity by both the HI and VN assays was observed in 24.3% (146/600) samples against strain D/660 and in 5.3% (32/600) samples against strain D/OK (Table 3, Figure 3). Reactivity to both IDV strains was observed in 3.7% (22/600) of samples by HI and in 19.2% (115/600) samples by VN (Table 3). In 3.2% (19/600) of serum samples, Abs for both IDV strains were detected by both assays (Table 3).

## 4. Discussion

Overall, 28.0% (168/600) of serum samples tested positive to either IDV strain D/660 or strain D/OK by HI. Abs specific to strain D/660 were detected in 135 samples from all 9 herds in the Ragusa prefecture and in 4 out of 6 herds in the Syracuse prefecture. Abs specific to strain D/OK were detected in 11 samples from 4 out of 9 herds in the Ragusa prefecture and 3 out of 6 herds in the Syracuse prefecture. These findings indicate that IDV circulates in the southeastern areas of the Sicilian region. Serological data for IDV in sheep in Italy are also available in a completely different ecosystem and climatic area, in the Po Valley, Northern Italy, from a 2016–2017 study [31]. A seroprevalence for IDV as high as 6.3% was observed by HI when screening a cohort of 506 sheep sera [16]. In a larger investigation in France spanning the years 2014–2018, over 1400 sheep sera and 600 goat sera from different regions of France were tested for the presence of IDV-specific Abs for IDV, revealing a seroprevalence of 0.5% in sheep and 3.2% in goats [10]. In Ireland, a seroprevalence for IDV of 4.5% was observed when screening of 288 sheep sera collected between 2016 and 2017 [9].

Serological studies to investigate the presence of IDV in small ruminants have also been performed in extra-European countries. In the USA and Canada, the presence of Abs for IDV has been found in 5.2% of 557 ovine sera and in 8.8% of 91 caprine sera [11]. The highest seroprevalence for IDV, 33.8%, has been reported in a small survey (80 goats) in China [26]. In an African study, the seroprevalence for IDV was 2.2% (2/135) and 1.4% (3/205) for sheep and goats, respectively, collected in Togo between 1991 and 2015. However, none of the 67 sheep and 34 goats from Benin had HI antibodies [24]. Studies conducted between 2017 and 2020 in Togo and Côte d’Ivoire reported low seroprevalence rates ranging from 2% (8/392) to 4.1% (7/171) in sheep and from 3.7% (6/163) to 4.4% (36/817) in goats [25].

Interestingly, in our study, we observed a marked difference in terms of seropositivity against the two IDV strains by HI. Most IDV-positive sera reacted against strain D/660, suggesting that this lineage was more common than the D/OK lineage in the Sicilian ovine population, thus mirroring the epidemiological data available for Europe and Italy [11,31]. Before 2017, all the Italian IDVs isolated belonged to the D/OK genetic cluster whilst the earliest D/660 strains were reported in Italy in 2018 in cattle imported from France [32].

The HI assay takes advantage of the ability of influenza viruses to bind with their surface receptor hemagglutinin (HA) to RBCs, forming macroscopic cell aggregates in a process termed hemagglutination [32]. In the presence of influenza-specific Abs blocking the RBC binding sites, hemagglutination is inhibited and this phenomenon can be exploited to reveal and quantify the serum HI antibodies [33]. Although the HI Abs are considered a good proxy of immunity to influenza viruses, only the neutralizing (VN) Abs are directly correlated with protective immunity [34]. All the ovine sera collected in this study were further tested by a VN assay and a total of 378 (63.0%) samples contained VN Abs either to strain D/660 or D/OK or to both. A total of 250 (41.6%) sera displayed VN Abs specific to strain D/660 whilst 13 (2.2%) sera had VN Abs specific to strain D/OK. Overall, the VN assay confirmed the results obtained by the HI assay, although the VN assay exhibited a higher sensitivity and the Ab titres were higher in VN than in HI. In addition, in some samples testing negative to both IDV strain D/OK and D/660 by HI, VN Abs could still be detected. Overall, studies based on HI might tend to underestimate the prevalence of IDV. Similar inconsistencies have been reported in studies conducted in small ruminants and humans [11,14]. In contrast, a study in horses in the USA reported a strong agreement between the HI and VN seroprevalence data [12].

Among the ovine sera tested in this study, seropositivity to both lineages was observed in 3.3% (22/600) samples in HI, and in 19.2% (115/600) samples in VN. Also, 3.2% (19/600) ovine sera tested positive for both IDV strains in HI and VN. As observed and suggested in other studies [11,12,14], in the case of sera with similar Ab reactivity against the two IDV lineages, we hypothesized exposure to the animal to an unidentified IDV strain able to induce cross-reactive Abs to both lineages. Alternatively, we explained this as the result of co-infection by more IDV strains, with strain D/660 being detected with higher prevalence in the ovine population, as observed in previous reports in Italy and Europe [31]. On the other hand, the finding of samples reacting against only one of the two strains could be explained by the possibility of the exposure to a D/660 IDV strain with mutations in key residues, abolishing epitopes in common with strain D/OK or, vice versa, exposure to a D/OK- like virus unable to cross-react with strain D/660 [12].

This study presented some limitations. It is known that cattle may play an important role in the spread of IDV, acting as a source of infection for other farm animals, wild animals, and eventually for humans [23]. Accordingly, the lack of seroprevalence studies of IDV in cattle and swine in Sicily does not offer a context to decipher the prevalence data obtained in the ovine population. Also, the lack of metadata (i.e., demographic and health status information) for the tested animals prevented further data mining. Finally, these data were generated only on two Sicilian prefectures encompassing the southeastern corner of Sicily and the seroprevalence results could be biased by geographical or temporal variations. However, the literature on IDV in sheep is thus far based mostly on analyses of small to medium archival collections of sera, thus hindering the extraction of deeper layers of information.

## 5. Conclusions

Overall, these findings provide evidence that multiple lineages of IDV infect sheep in Sicily, Italy. The prevalence rates for IDV in the investigated areas were markedly higher than those reported in the literature. It will be important to increase IDV surveillance in ovine populations, in larger structured studies, and for prolonged periods. Also, it will be important to extend surveillance for IDV to other livestock and wildlife animals, to unveil the ecology of IDV, and to understand the reasons for unusual epidemiological patterns. Finally, since IDV has a large host range and antibodies to IDV have been identified in humans, these studies will help assess, if any, the zoonotic risks posed by IDV.

## Figures and Tables

**Figure 1 pathogens-13-00162-f001:**
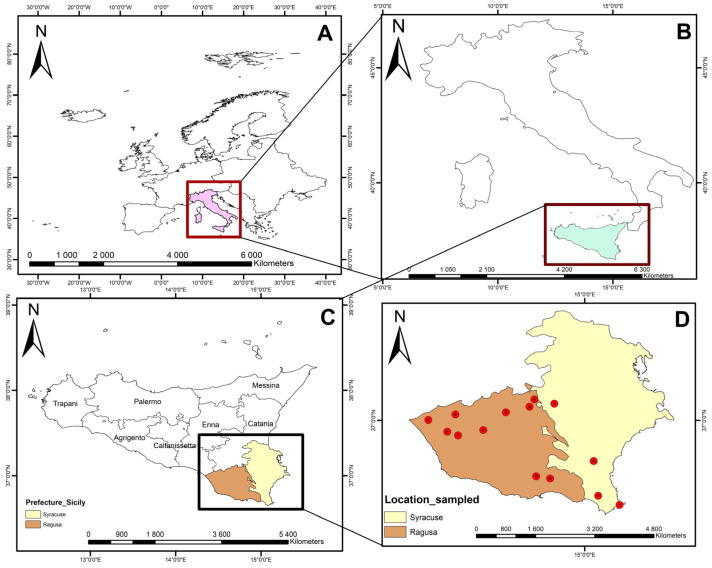
Map of the study area within the European continent (**A**), the Italian country (**B**), the Sicily boundaries in Italy, (**C**) and the geographic distribution of dairy sheep herds in Ragusa and Syracuse prefectures (**D**).

**Figure 2 pathogens-13-00162-f002:**
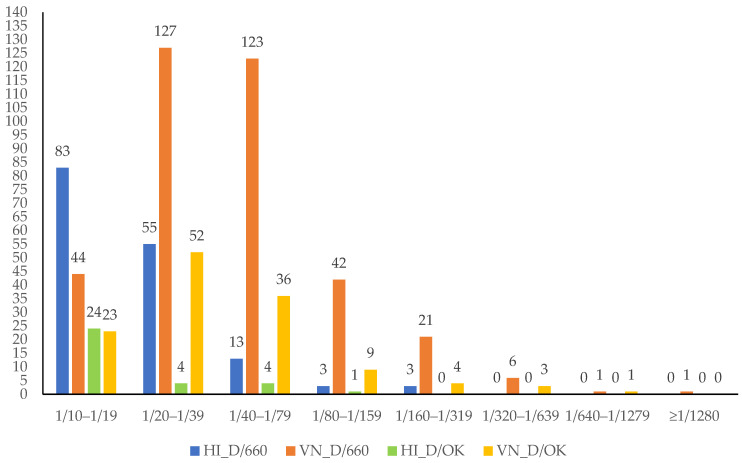
Distribution of influenza D virus (IDV) positive antibody in ovine samples. The haemagglutination inhibition (HI) HI and virus neutralization (VN) VN titres ≥ 10 were used as the cut-off value for seropositive samples. Regarding the reference strain influenza D/bovine/Oklahoma/660/2013 (D/660), the highest positive rate HI titre was in the range 1/160 to 1/319 whilst the highest positive rate VN titre was ≥1280. Regarding the reference strain influenza D/swine/Italy/199724/2015 (D/OK), the highest positive rate HI titre was in the range 1/80 to 1/159 whilst the highest positive rate VN titre was in the range 1/640-1/1279.

**Figure 3 pathogens-13-00162-f003:**
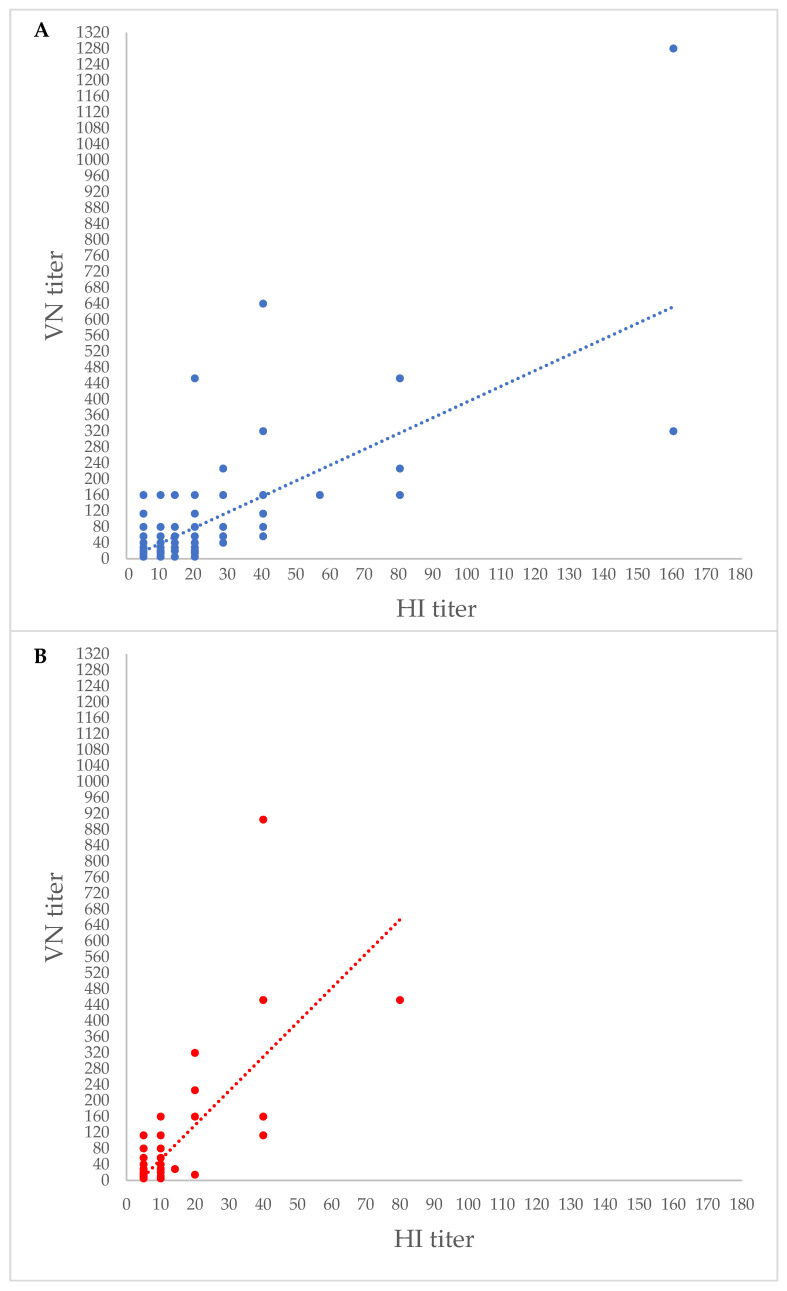
Scatter plots of Abs titres obtained by haemagglutination inhibition (HI) and virus neutralization (VN) assays. Abs to influenza D/bovine/Oklahoma/660/2013 (D/660) (**A**) and influenza D/swine/Italy/199724/2015 (D/OK) (**B**) strains were assessed.

**Table 1 pathogens-13-00162-t001:** Information on geographic location of serum samples collected from dairy sheep herds in Sicily, Southern Italy.

Prefecture	Sheep Herd	Location and Zip Code	Samples Collected
RG	A	Acate 97011,	50
	D	Comiso, 97013,	65
	E	Baucino, Vittoria, 97019	64
	F	Niscescia, Vittoria, 97019	43
	I	Linfante, Vittoria, 97019	70
	K	Ispica, 97014	50
	L	Giarratana, 97010	35
	M	Chiaramonte Gulfi, 97012	20
	O	Modica, 97015	10
Total RG	#9		407
SR	B	Buscemi, 96010	57
	C	Vallefame, 96010	7
	G	Noto, 96017	57
	H	Pachino, 96018	12
	J	Palazzolo Acreide, 96010	45
	N	Portopalo di Capo Passero, 96010	15
Total SR	#6		193
Total	#15		600

# number.

**Table 2 pathogens-13-00162-t002:** Distribution of samples with positive (≥1/10) and negative (<1/10) Abs to influenza D/bovine/Oklahoma/660/2013 (D/660) and influenza D/swine/Italy/199724/2015 (D/OK) strains assessed by haemagglutination inhibition (HI) and virus neutralization (VN) assays.

Prefecture	Sheep Herd	Samples Collected	HI_D/660 Titre		VN_ D/660 Titre		HI_ D/OK Titre		VN_ D/OK Titre	
			≥1/10	<1/10	≥1/10	<1/10	≥1/10	<1/10	≥1/10	<1/10
RG	A	50	12	38	32	18	4	46	7	43
	D	65	8	57	34	31	1	64	4	61
	E	64	22	42	37	27	5	59	9	55
	F	43	11	32	16	27	2	41	4	39
	I	70	29	41	39	31	1	69	15	55
	K	50	21	29	41	9	1	49	16	34
	L	35	7	28	23	12	2	33	11	24
	M	20	2	18	13	7	0	20	5	15
	O	10	7	3	8	2	4	6	5	5
Total RG	#9	407	119	288	243	164	20	387	76	331
SR	B	57	3	54	43	14	2	55	9	48
	C	7	0	7	6	1	0	7	0	7
	G	57	10	47	30	27	5	52	11	46
	H	12	0	12	8	4	2	10	4	8
	J	45	20	25	26	19	4	41	23	22
	N	15	5	10	9	6	0	15	5	10
Total SR	#6	193	38	155	122	71	13	180	52	141
Total	#15	600	157	443	365	235	33	567	128	472

RG/Ragusa; SR/Syracuse; # number.

**Table 3 pathogens-13-00162-t003:** Antibodies against influenza D/bovine/Oklahoma/660/2013 (D/660) and influenza D/swine/Italy/199724/2015 (D/OK) strains consistently observed by haemagglutination inhibition (HI) and/or virus neutralization (VN) assays.

Prefecture	Sheep Herd	HI Assay		VN Assay		D/660		D/OK		HI and VN Assays	
		D/660	D/OK	D/660	D/OK	HI Assay	VN Assay	HI Assay	VN Assay	D/660	D/OK
RG	A	4		6		9		3		2	
	D	0		4		8		1		0	
	E	4		8		21		5		3	
	F	2		4		11		2		2	
	I	0		13		27		1		0	
	K	1		16		19		1		1	
	L	0		9		7		2		0	
	M	0		4		2		0		0	
	O	4		5		7		4		4	
Total RG	#9	15		69		111		19		12	
SR	B	0		9		3		2		0	
	C	0		0		0		0		0	
	G	3		9		10		5		3	
	H	0		3		0		2		0	
	J	4		20		18		4		4	
	N	0		5		4		0		0	
Total SR	#6	7		46		35		13		7	
Total	#15	22		115		146		32		19	

## Data Availability

The original contributions presented in the study are included in the article; further inquiries can be directed to the corresponding author.

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
