# Peer review of "Serological Evidence for Circulation of Influenza D Virus in the Ovine Population in Italy"

_pathogens, 2024, doi:10.3390/pathogens13020162_

Round 1
Reviewer 1 Report
Comments and Suggestions for Authors
The manuscript by Gianovito Lanave et.al. presents a serological survey of Influenza D virus (IDV) in ovine serum samples from the year 2022 in Sicily, Southern Italy. Six hundred serum samples were tested by haemagglutination inhibition (HI) and virus neutralization (VN) assays with reference strains, D/660 and D/OK, representing two distinct IDV lineages circulating in Italy. The study found that 168 (28.0%) tested positive for either IDV strain D/660 or D/OK or to both by HI, while 378 (63.0%) tested positive to either IDV strain D/660 or D/OK or to both by VN. These results demonstrate the circulation of IDV in ovine dairy herds in Sicily. It is a screening and informative study.
However, I have following questions about this manuscript:
1. Why are the numbers of positive results in the HI and VN assays so different? The VN assay positive count is more than twice that of the HI assay positive count (line 29 to 31). Is there a good explanation for this difference? What are the distinctive characteristics between the serum group with both HI and VN positive results and the group with only HI or VH positive result?
2. Line 114 indicates that the HI assay was performed “in triplicated”. What are the criteria for considering a sample positive in the HI assay when performed in triplicate? Must it be positive in all the 3 tests? How did the authors handle samples with inconsistent results in the triplicated HI assay?
3. In the section of VN assay (line 119 to 128), the authors did not mention whether there were repeats of the VN assay for each sample. Was the VN assay performed only once for each sample?
4. For the low-titer serum samples (marginal positive calls), VN assay from purified IgG from them are necessary for confirmation.
5. Raw results of the HI and VN assays should be included in supplemental tables for review.
Author Response
Dear Referee,
herein you can find a point-by-point response to the your comments for the manuscript pathogens-2846475 entitled “Serological evidence for circulation of influenza D virus in the ovine population in Italy” submitted to Pathogens.
Thanks in advance
Best regards
Gianvito Lanave
The manuscript by Lanave et.al. presents a serological survey of Influenza D virus (IDV) in ovine serum samples from the year 2022 in Sicily, Southern Italy. Six hundred serum samples were tested by haemagglutination inhibition (HI) and virus neutralization (VN) assays with reference strains, D/660 and D/OK, representing two distinct IDV lineages circulating in Italy. The study found that 168 (28.0%) tested positive for either IDV strain D/660 or D/OK or to both by HI, while 378 (63.0%) tested positive to either IDV strain D/660 or D/OK or to both by VN. These results demonstrate the circulation of IDV in ovine dairy herds in Sicily. It is a screening and informative study.
However, I have following questions about this manuscript:
R1.1. Why are the numbers of positive results in the HI and VN assays so different? The VN assay positive count is more than twice that of the HI assay positive count (line 29 to 31). Is there a good explanation for this difference? What are the distinctive characteristics between the serum group with both HI and VN positive results and the group with only HI or VN positive result?
Reply to R1.1. We agree with the referee that these patterns can be odd/curious and can elicit some reflections in the readers. The possible reason for the difference in HI and VN titers relies mainly on the type of antibodies assessed in the two different assays. VN is usually a more sensitive assay since it detects functional antibodies (blocking the receptor controlling virus entry), and at the same time, it is regarded as a highly specific assay. HI is used as a good proxy of VN but the assay does not measure directly VN antibodies. Also, the results can be confounded by other factors. For instance, the reactivity of VN and HI antibodies to different strains of influenza can vary quantitatively. One strain can be recognized to a higher extent (higher VN titers) than other strains, based on the antigenic/genetic relatedness with the actual virus strain that elicited the immune response. For this reason, we used two IDV strains in our investigation. These variations can be also seen in HI. Additionally, since the antibodies decline over time, the VN or HI antibodies in some sera can go under the threshold of detection for one antigen, but can still be detectable using another antigen. Finally, it is possible that in some sera antibodies can be still revealed by HI, but not by VN, due to antigenic differences that in theory should affect more the VN assay than the HI assay. Yet these possible explanations are rather speculative.
Overall, based on the VN results, the seroprevalence provided by the HI assay may be slightly underestimated. The same inconsistencies were also observed in previous studies (Trombetta et al., 2022, Quast et al., 2015, Nedland et al., 2018). However, this does not affect the general picture of the epidemiology of a pathogen, i.e., in this case of IDV, obtained using serological tools. We evidenced in the manuscript at page 9, line 269-270 that there was a higher reactivity against one antigen. Also, on page 9, lines 275-291, we commented on the different reactivities observed in VN and HI. We also remarked in the text (lines 285-287) that “the VN assay confirmed the results obtained by the HI assay, although the VN assay exhibited a higher sensitivity and the Ab titers were higher in VN than in HI.”
R1.2. Line 114 indicates that the HI assay was performed “in triplicated”. What are the criteria for considering a sample positive in the HI assay when performed in triplicate? Must it be positive in all the 3 tests? How did the authors handle samples with inconsistent results in the triplicated HI assay?
Reply to R1.2. The cutoff of antibody titer for the HI and the VN assays was set to 1/10 for each experiment performed as stated in the manuscript (see lines 150-152 pag 5). We calculated the average antibody titre after performing the experiments in triplicate. The positive and negative results in each test for the 600 sera were consistent in all three experiments performed.
R1.3. In the section of VN assay (line 119 to 128), the authors did not mention whether there were repeats of the VN assay for each sample. Was the VN assay performed only once for each sample?
Reply to R1.3. We performed the VN assay in triplicate. Accordingly, we corrected this information in the text (see line 127 page 4).
R1.4. For the low-titer serum samples (marginal positive calls), VN assay from purified IgG from them are necessary for confirmation.
Reply to R1.4.
Purification of IgG is not a standard procedure in serology, to our knowledge. Usually, the sera are pre-treated to remove aspecific hemagglutinins before HI. In our experiments, we pretreated the sera with RDE (receptor destroying enzyme) and thereafter the sera were absorbed on turkey red blood cells. The virus neutralization assay with live virus, as performed in this study, is considered the “gold standard” for detection of antibodies for influenza virus infection and/or vaccination. Moreover, the VN assay is significantly more sensitive than HI assay for detecting low-titer seroconversion. All serum samples were heat-inactivated at 56°C for 30 minutes for complement inactivation, useful for decreasing the number of false-positive results. Overall, as stated above, purification of IgG is not a standard procedure in serology, and we do not have in our laboratory the kits and equipment for this purification step. Please consider that all the experiments were done on triplicates in VN and HI, consuming the serum samples. Also, provided that this is a cost-effective solution, we are not sure that this treatment would add relevant info to the general picture, justifying the additional costs and work. Yet, we agree that for other applications this purification could be useful, as stated by R1.
R1.5. Raw results of the HI and VN assays should be included in supplemental tables for review.
Reply to R1.5. A table with raw results of the HI and VN assays were provided as supplemental material (Appendix A).
Reviewer 2 Report
Comments and Suggestions for Authors
Comments for the Authors
The Communication manuscript entitled "Serological evidence for circulation of influenza D virus in the ovine population in Italy" is a well written study aiming at assessing the prevalence of antibodies against IDV (two lineage representative strains: D/660 and D/OK) in ovine serum samples in Sicily, Southern Italy. The results showed that 157 (157 of 600, 26.2%) samples were positive for antibodies against D/660, and 33 (33 of 600, 5.5%) samples were positive for antibodies against D/OK by HI assays. When these samples were examined by VN assays, 365 (365 of 600, 60.8%) samples were positive for antibodies against D/660, and 128 (128 of 600, 21.3%) samples were positive for antibodies against D/OK. These findings clearly demonstrate that IDV exposes to ovine dairy herds in Sicily. Here, I have some questions and suggestions:
Major
1, In this study, it is interesting that there is a significant difference (26.2% vs 5.5%) in terms of seropositivity against the D/660 and D/OK strains by HI assays. To well support these results, the reviewer would suggest the authors to present the HI cross-reactivity between the two IDV lineage-representative strains by using their specific hyperimmune sera, and titers (TCID50 and HA) of viral stocks used for HI and VN assays.
2, In this study, inconsistencies of results from the HI and VN assays have been reported by the authors. In addition, the overwhelming majority of seropositive samples presented by the authors show titers (HI and VN) less than 40. For these reasons, the reviewer would suggest the authors to further validate assay specificity by examining a subset of these positive samples (from low to high titers) via HI and VN assays with other types of influenza viruses.
3, Considering differences between seropositivity against D/660 and D/OK and the inconsistent results between the HI and VN assays, the reviewer suggests the authors to generate scatter plots (refer to Ref.12, by Nedland, H., et al., Fig.1) showing HI and VN antibody titers between the two strains.
Minor
4, In the abstract, lines 29-31, the authors describe that “Out of 600 tested samples, 168 (28.0%) tested positive to either IDV strain D/660 or D/OK or to both by HI whilst 378 (63.0%) tested positive to either IDV strain D/660 or D/OK or to both by VN”. The reviewer suggests the authors to present the results in detail.
5, In the introduction, lines 56-57, the authors describe that “Recently, an additional lineage, D/Bursa2013, has been reported in Europe”. Since the phylogenetic analysis in the cited study includes a very short HEF sequence (Accession: OM639976.1, 472bp), in the opinion of the reviewer, it may be less rigorous to consider D/Bursa2013 as an additional IDV lineage. Please also confirm it.
Author Response
Dear Referee,
herein you can find a point-by-point response to the your comments for the manuscript pathogens-2846475 entitled “Serological evidence for circulation of influenza D virus in the ovine population in Italy” submitted to Pathogens.
Thanks in advance
Best regards
Gianvito Lanave
The Communication manuscript entitled "Serological evidence for circulation of influenza D virus in the ovine population in Italy" is a well written study aiming at assessing the prevalence of antibodies against IDV (two lineage representative strains: D/660 and D/OK) in ovine serum samples in Sicily, Southern Italy. The results showed that 157 (157 of 600, 26.2%) samples were positive for antibodies against D/660, and 33 (33 of 600, 5.5%) samples were positive for antibodies against D/OK by HI assays. When these samples were examined by VN assays, 365 (365 of 600, 60.8%) samples were positive for antibodies against D/660, and 128 (128 of 600, 21.3%) samples were positive for antibodies against D/OK. These findings clearly demonstrate that IDV exposes to ovine dairy herds in Sicily. Here, I have some questions and suggestions:
Major
R2.1. In this study, it is interesting that there is a significant difference (26.2% vs 5.5%) in terms of seropositivity against the D/660 and D/OK strains by HI assays. To well support these results, the reviewer would suggest the authors to present the HI cross-reactivity between the two IDV lineage-representative strains by using their specific hyperimmune sera, and titers (TCID50 and HA) of viral stocks used for HI and VN assays.
Reply to R.2.1. The referee R2 suggests making cross-evaluation assays in HA and VN using the two IDV variants and specific hyperimmune antisera. This has already been described elsewhere (Nedland et al., 2018; doi 10.1111/zph.12423. Huang et al., 2021; doi: 10.1080/22221751.2021.1910078). As specified in M&M section (lines 102-103 page 3), we used a specific hyperimmune serum sample for D/OK lineage only. Due to Italian regulations, we were unable to import specific hyperimmune serum for D/660. We have added the titers of D/OK specific hyperimmune sample to both IDVs lines 142-144 page 4).
R2.2. In this study, inconsistencies of results from the HI and VN assays have been reported by the authors. In addition, the overwhelming majority of seropositive samples presented by the authors show titers (HI and VN) less than 40. For these reasons, the reviewer would suggest the authors to further validate assay specificity by examining a subset of these positive samples (from low to high titers) via HI and VN assays with other types of influenza viruses.
Reply to R2.2. We have replied to a similar question posed by referee R1 (see reply to query R1.1) Overall, based on the serological results, the seroprevalence data provided by the HI assay may be slightly underestimated. The same inconsistencies were also observed in previous studies (Trombetta et al., 2022, Quast et al., 2015, Nedland et al., 2018). However, we would like to point out that, in the majority of other studies, the sera have been screened only in HI, and these possible discrepancies were not observed. In a previous study (Trombetta et al., 2022; doi: 10.1002/jmv.27466) we tested hyperimmune samples to IAV, IBV, ICV and IDVs for D/OK and D/660 by HI assay showing positive results only for IDVs. The positivity cut-off of the HI assay for IDV has been established (titre=10) elsewhere by Saegerman et al., 2020. We think that testing the sera with additional assays or antigens would not generate exhaustive and resolutive information.
R2.3. Considering differences between seropositivity against D/660 and D/OK and the inconsistent results between the HI and VN assays, the reviewer suggests the authors to generate scatter plots (refer to Ref.12, by Nedland, H., et al., Fig.1) showing HI and VN antibody titers between the two strains.
Reply to R2.3. Scatter plots were generated as suggested and provided in the manuscript as figure 3
Minor
R2.4. In the abstract, lines 29-31, the authors describe that “Out of 600 tested samples, 168 (28.0%) tested positive to either IDV strain D/660 or D/OK or to both by HI whilst 378 (63.0%) tested positive to either IDV strain D/660 or D/OK or to both by VN”. The reviewer suggests the authors to present the results in detail.
Reply to R2.4. As requested also by referee 1, a table with raw results of the HI and VN assays were provided as supplemental material (Appendix A).
R2.5. In the introduction, lines 56-57, the authors describe that “Recently, an additional lineage, D/Bursa2013, has been reported in Europe”. Since the phylogenetic analysis in the cited study includes a very short HEF sequence (Accession: OM639976.1, 472bp), in the opinion of the reviewer, it may be less rigorous to consider D/Bursa2013 as an additional IDV lineage. Please also confirm it.
Reply to R2.5. We agree with referee R2’s observation. Accordingly, in the text, we modified the sentence to “Recently, an additional lineage, D/Bovine/Turkey-Bursa/ET-130/2013, has been tentatively proposed, although phylogenetic analysis relied solely on a partial HEF gene sequence (see lines 57-59 pag 2).
Reviewer 3 Report
Comments and Suggestions for Authors
In their article” Serological evidence for circulation of influenza D virus in the ovine population in Italy “ the authors present a study about seroprevalence of antibodies against IDV in small ruminants in Sicily, Southern Italy. Influenza D virus is a novel virus with zoonic potential. That is why it is very important to investigate and collect information about it, about susceptible species and etc. This article gives us a useful epidemiological data about the spared of the virus in ovine population in Sicily.
In my opinion, the manuscript needs some adjustments.
Introduction. Lines 54-59. Write the whole name of the viruses, for example D/swine/Oklahoma/1334/2011.
Materials and Methods. Figure 1. is better to be in Section Results.
Results. Line 138. The first sentence is not proper. There are many repetitions of “600 samples”, revise them.
Discussion. This part needs a serious revision.
Lines 207-211. The first paragraph is repetition of the text from Results.
Line 220. You need to cite.
Line 226. The sentence needs revision.
Lines 241-247. The text does not proper for this part, there is no relation with provided information.
In conclusion, I will say that the provided data is useful and interesting, but needs some revision.
Author Response
Dear Editor,
herein you can find a point-by-point response to the your comments for the manuscript pathogens-2846475 entitled “Serological evidence for circulation of influenza D virus in the ovine population in Italy” submitted to Pathogens.
Thanks in advance
Best regards
Gianvito Lanave
In their article” Serological evidence for circulation of influenza D virus in the ovine population in Italy “the authors present a study about seroprevalence of antibodies against IDV in small ruminants in Sicily, Southern Italy. Influenza D virus is a novel virus with zoonic potential. That is why it is very important to investigate and collect information about it, about susceptible species and etc. This article gives us a useful epidemiological data about the spared of the virus in ovine population in Sicily.
In my opinion, the manuscript needs some adjustments.
R3.1. Introduction. Lines 54-59. Write the whole name of the viruses, for example D/swine/Oklahoma/1334/2011.
Reply to R3.1. The referee R3 is correct. D/swine/Oklahoma/1334/2011 is the prototype strain of the lineage, shortened to D/OK for the sake of simplicity. We clarified this in the introductory part of the MS, reporting the full names of the strains (see lines 54-59, page 2).
R3.2. Materials and Methods. Figure 1. is better to be in Section Results.
Reply to R3.2. We agree with referee R3 that Figure 1 would be better located in Section Results. Accordingly, we moved Figure 1 at the beginning of the section results.
R3.3. Results. Line 138. The first sentence is not proper. There are many repetitions of “600 samples”, revise them.
Reply to R3.3. The sentence was moved and integrated in the text (see lines 64-67, page 2), as suggested by the referee R3. As for the repetitions, we checked the MS and we avoided, when possible, repetitions.
R.3.4. Discussion. This part needs a serious revision.
Reply to R.3.4. We thank the referee for the suggestions that improved the quality of the manuscript. We modified the discussion according to your suggestions.
R.3.5. Lines 207-211. The first paragraph is repetition of the text from Results.
Reply to R.3.5. We removed the whole paragraph, as suggested.
R.3.6. Line 220. You need to cite.
Reply to R.3.6. A reference (#31) was cited in the text (see line 253, page 9), as suggested.
R.3.7. Line 226. The sentence needs revision.
Reply to R.3.7. The sentence was modified, as follows “Serological studies to investigate the presence of IDV in small ruminants have been performed also in extra-European countries” (see lines 259-260, pag 9).
R.3.8. Lines 241-247. The text does not proper for this part, there is no relation with provided information.
Reply to R.3.8. We added this section, since there are differences between HI and VN in terms of results, due to the different targets of the two assays (see lines 275-281, page 10). In several studies, only HI has been used. However, in our study we tested the sera in parallel, using both the assays, and obtaining some slightly different results. Accordingly, we wanted to provide a background to the readers. See our reply to the referee R1 (reply to query R1.1)
R.3.9. In conclusion, I will say that the provided data is useful and interesting, but needs some revision.
Reply to R.3.9. We thank the referee for the appreciation of the manuscript. We modified the text following his pieces of advice.